# Multi-Task Learning for Scene Text Image Super-Resolution with Multiple Transformers

Kosuke Honda [1], Masaki Kurematsu [1], Hamido Fujita [2,3,4,5,*] and Ali Selamat [3]

1 Faculty of Software and Information Science, Iwate Prefectural University, Takizawa 020-0693, Japan
2 Faculty of Information Technology, HUTECH University, Ho Chi Minh City 72308, Vietnam
3 Malaysia-Japan International Institute of Technology (MJIIT), Universiti Teknologi Malaysia, Kuala Lumpur 54100, Malaysia
4 Regional Research Center, Iwate Prefectural University, Takizawa 020-0693, Japan
5 i-SOMET Inc., Morioka 020-0104, Japan
* Correspondence: h.fujita@hutech.edu.vn or fujitahamido@utm.my or hfujita-799@acm.org

**Abstract:** Scene text image super-resolution aims to improve readability by recovering text shapes from low-resolution degraded text images. Although recent developments in deep learning have greatly improved super-resolution (SR) techniques, recovering text images with irregular shapes, heavy noise, and blurriness is still challenging. This is because networks with Convolutional Neural Network (CNN)-based backbones cannot sufficiently capture the global long-range correlations of text images or detailed sequential information about the text structure. In order to address this issue, this paper proposes a Multi-task learning-based Text Super-resolution (MTSR) Network to approach this problem. MTSR is a multi-task architecture for image reconstruction and SR. It uses transformer-based modules to transfer complementary features of the reconstruction model, such as noise removal capability and text structure information, to the SR model. In addition, another transformer-based module using 2D positional encoding is used to handle irregular deformations of the text. The feature maps generated from these two transformer-based modules are fused to attempt improvement of the visual quality of images with heavy noise, blurriness, and irregular deformations. Experimental results on the TextZoom dataset and several scene text recognition benchmarks show that our MTSR significantly improves the accuracy of existing text recognizers.

**Keywords:** scene text image super-resolution; multi-task learning; scene text recognition; transformer; attention mechanism

## 1. Introduction

Today, information in images is essential in various situations, including daily life, business scenes, and medical settings. However, the information is often lost due to several factors, such as complex backgrounds, noise, blurriness, and low-resolution images. In recent years, there have been many studies to obtain high visual quality images without noise, including approaches in bio-imaging [1,2] using a scientific camera such as CMOS and approaches using super-resolution (SR) processing [3–6]. In scene text recognition (STR) which we address in this study, the visual quality of images is also an important factor because recognition performance is significantly affected by them. STR aims to convert text images into computer-readable and editable symbols, a fundamental and vital task in computer vision. This technique has been widely applied to studies for automation or efficiency, such as license plate recognition, text retrieval, and ID card recognition [7–9]. The performance of STR has improved significantly in recent years with the development of deep learning. However, images with various quality degradations, such as low resolution (LR) or blurred structures, cause significant difficulties in text recognition. Against this background, recently, there have been many studies to improve the resolution and visual quality of text images for STR.

Single image super-resolution (SISR) is a fundamental task in computer vision that reconstructs fine detail from a degraded LR image and generates a high-resolution (HR) image. Similar to STR, SISR techniques have also improved significantly with the development of deep learning. Researchers have developed many SISR methods for scene text images in the past few years, as the task named scene text image super-resolution (STISR). Several studies [10,11] adopt the deep learning-based method designed for generic SISR to STISR, proving the effectiveness of using super-resolution methods as preprocessing for STR. However, these methods are insufficient for text images with heavy quality degradation since they can not distinguish between generic and text images and take into account the specific properties of text images. Then, the deep learning based-methods specific to text images [12–15] were developed, providing significant performance improvements.

Most STISR methods [12–15] specific to text images train using TextZoom [12]. TextZoom is the first dataset for STISR containing pairs of degraded text images with high-quality text images. A representative STISR method of TSRN [12] captures sequential features in text images using Bidirectional LSTM (BLSTM) [16] and text-specific loss using a gradient profile. Another representative method, TBSRN [15], attempts to restore the text shapes using pre-recognized text semantic information. However, despite these technical advances, restoring text images that are heavily blurred or noisy, have crooked or distorted text, or used unique fonts is still challenging. One of the main reasons for this is that networks using CNN-based backbones cannot sufficiently capture global long-range correlations in text images and detailed sequential information about text structure.

In this paper, we propose a new STISR network, termed Multi-task learning based Text SR (MTSR) Network, which employs multi-task architecture for SR and image reconstruction to approach this problem. Image reconstruction is a close task to SR in computer vision, which generates a high-visual-quality image of the same pixels from an input image. The models of this task tend to have more robust noise-removal capability and information on the correct text structure than SR models. Therefore, we employed multi-task learning (MTL) architecture for image reconstruction and SR to take advantage of these properties. Our MTSR consists of the SR and Reconstruction branches, which simultaneously train end-to-end. The features extracted in the reconstruction branch are sent SR branch for feature representation sharing/transfer using the Feature Sharing Transformer (FST). In addition, we use a backbone in the SR branch containing the Feature Enhancement Transformer (FET) with adaptive 2D positional encoding and multi-head attention (MHA) for capturing global long-range correlations and sequential spatial information in arbitrary orientation. Features from these two transformers are fused and output as a final feature map for SR. In this work, we employ two types of 2D positional encoding for FET, such as Absolute Positional Encoding (APE) and Relative Positional Encoding (RPE), to analyze the effectiveness of these by comparison. Examples of generated SR images using MTSR and the predicted results are shown in Figure 1. The main contributions of our work are as follows:

- We propose the MTSR network that simultaneously performs text image reconstruction and super-resolution using two transformers to recover text shapes.
- The FST transfers the complementary features extracted from the reconstruction branch as an attention map to the SR branch for obtaining robust noise-removal capability and information on the correct text structure. Furthermore, the FET in the SR branch captures global long-range correlations and sequential spatial information in arbitrary orientation by adaptive 2D PE and MHA.
- Evaluation results on the TextZoom dataset [12] and popular STR benchmarks [17–20] prove the effectiveness of the proposed method. In addition, the proposed model exhibits outstanding generalization performance and learning efficiency due to the MTL of image reconstruction and super-resolution.

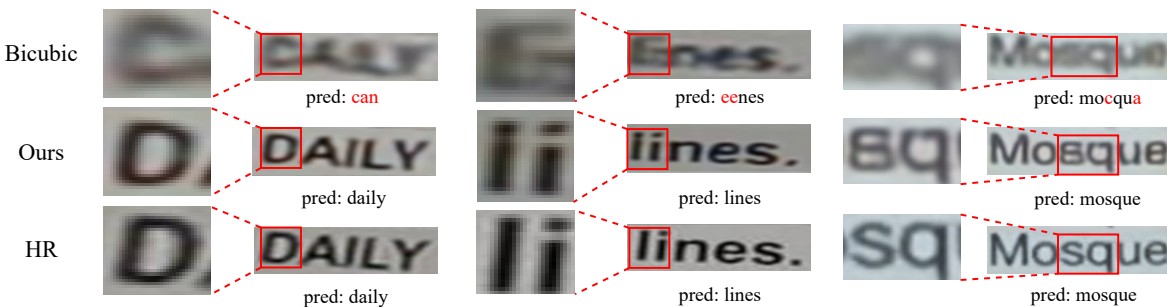

**Figure 1.** Examples of SR images generated using our MTSR and prediction using ASTER [21] on the TextZoom [12]. Red texts mean the results of misrecognition.

## 2. Related Works

### 2.1. Single Image Super-Resolution (SISR)

SISR aims to reconstruct fine details from degraded LR images to generate HR images. Most recent deep learning-based SISR methods use Convolutional Neural Networks (CNNs) as the backbone and achieve significantly improved performance compared to earlier work. SRCNN [3], with its simple structure using a three-layer CNN, is the leading general-purpose SISR method. Since its proposal, more complex and higher-performance SISR architectures have been developed. As other computer vision tasks have developed, SISR methods have been developed that employ Residual blocks [4], attention mechanisms [5], generative adversarial networks [6], and transformer [22]. Among the SISR methods, TTSR [22], a reference-based SR method, uses transformers for texture transfer between LR and reference images and has achieved significant performance gains.

### 2.2. Scene Text Recognition (STR)

Recently, scene text recognition techniques have made significant progress with the development of deep learning. Some traditional and early approaches [20,23,24] recognize text bottom-up, from character to word and word to text. However, this approach needs to be revised for Improved readability of scene text images because these methods cannot capture sequential information in the text image. CRNN [25] combines CNNs and the Recurrent Neural Network (RNN) to capture semantic and sequential information from the whole text image. In ASTER [21], the input images are rectified using Spatial Transformer Network (STN) [26], and then the feature maps are enhanced using the attention mechanism. Furthermore, advanced methods, such as Attention-based methods in the 2D direction [27] and Transformer-based STR [28], have significantly improved STR benchmarks in recent years. However, although recent significant improvements in performance, it is still challenging to recognize LR text images.

### 2.3. Scene Text Image Super-Resolution (STISR)

STISR is an SR task specific to scene text images. Its main difference from generic SR for natural scene images is that it aims not only to improve resolution and visual quality but also to recover the shape of the text and reconstruct the LR text images to the computer and human-recognizable images. Most early approaches [10,11] to STISR have extended generic SR methods for text images. For example, Dong et al. employed SRCNN [3] as a backbone network for text SR and achieved state-of-the-art performance in the ICDAR 2015 benchmark. Another example is the approach in [11], which uses a Laplacian pyramid network as a backbone network to capture text details by fusing multiple features from the middle layer.

These early approaches used LR images downsampled from HR images using interpolation methods such as BICUBIC and Bilinear for the training dataset. On the other hand, TSRN [12] uses the dataset TextZoom, which contains pairs of LR and HR text images of real-world scenes, for training, resulting in a significant improvement in performance on real-world scenes. In addition, TSRN utilizes a backbone network with BLSTM [16] to

attempt to capture sequential horizontal and vertical information. Since TextZoom [12] was proposed, it has become the standard for STISR training datasets and has given rise to many follow-up studies. TSRGAN [14] is a GAN-based STISR network that improves the representativeness of feature maps by introducing triplet attention. Chen et al. [15] proposed a training method that uses transformer-based networks and text-specific loss functions to improve STISR performance. TPGSR [13] pre-recognizes the semantic information of text and utilizes it in STISR to recover semantically correct text images with convincing visual quality.

## 3. Methodology

### 3.1. Overall Architecture

In this paper, we propose the MTSR network for STISR that MTL architecture of SR and reconstruction. The overall architecture of our MTSR is shown in Figure 2. The input image is $h \times w \times 4$ ($h$ and $w$ are the height and width) shape concatenated with an RGB image and a binary mask following [12]. MTSR consists of an image reconstruction branch and an SR branch in parallel, each processing LR text images rectified by the Spatial Transformer Network (STN) [26]. First, a shallow feature map is extracted by $1 \times 1$ convolution in each branch, and then features are extracted by SRBlock and Reconstruction Block (RecBlock) based on EDSR [29] consisting of CNNs and BLSTM [16], shown in Figure 3. More details of SRBlock are introduced in Appendix A. The extracted features from the reconstruction branch are sent as a key and value to the FST module for sharing/transferring the complementary characteristics of the reconstruction branch to the SR branch. On the other hand, the features extracted in the SR branch are sent to two modules, the FET, which enhances the feature using MHA and adaptive 2D positional encoding, and FST. The feature maps generated from the two transformer modules are fused and output as the final feature map. The model is trained to minimize the loss function between the images generated from each branch and the target images. Here, the target image for the SR branch is a pre-prepared HR image in TextZoom [12], and the target image for the reconstruction branch is a high-quality small image downsampled from the HR image.

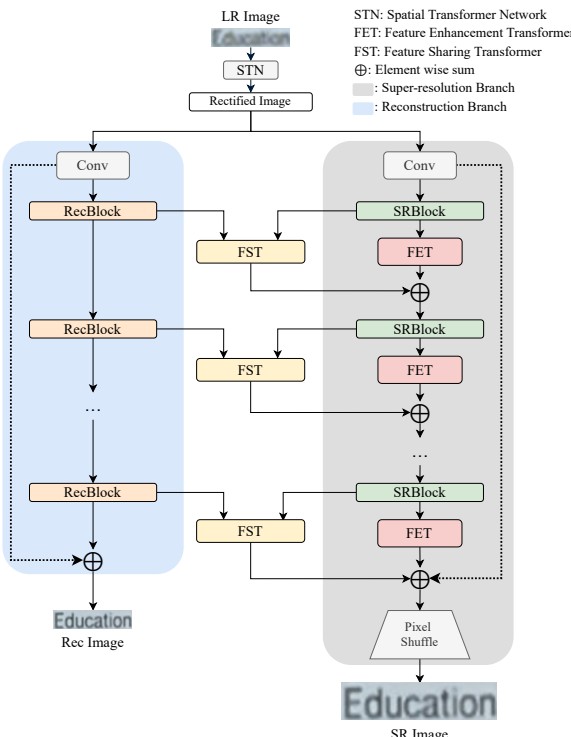

**Figure 2.** The overall architecture of our proposed MTSR Network for STISR. FET and FSR stand for Feature Enhancement Transformer and Feature Sharing Transformer, respectively.

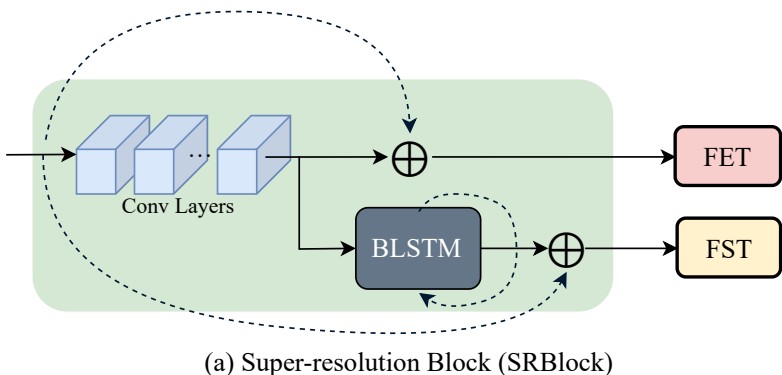

(a) Super-resolution Block (SRBlock)

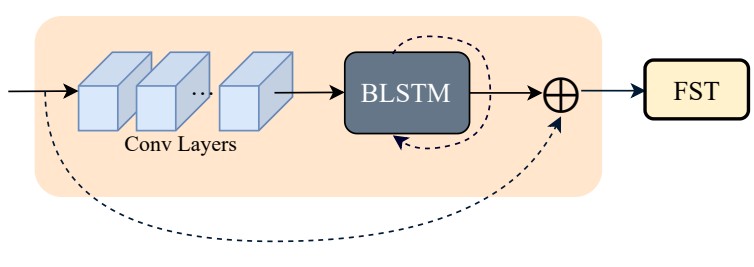

(b) Reconstruction Block (RecBlock)

**Figure 3.** SRBlocks and RecBlocks are feature extractors for the Reconstruction and SR branches, respectively. The Convolutional layers in the SRBlock and RecBlocks are both based on EDSR [29].

*3.2. Feature Sharing Transformer (FST) for MTL*

The reconstruction branch aims to remove noise from LR text images and generate visually high-quality images without up-sampling. The image reconstruction task is very close to SR in computer vision. However, it tends to be superior in removing noise and capturing structural features of objects because they clean the image without up-sampling. To exploit these characteristics for text reconstruction, we adopt the MTL architecture of SR and image reconstruction, and the features extracted reconstruction branch are shared/transmitted through the FST module to the SR branch. The architecture of the FST module is inspired by the transformer [22,30], which shares/transfers features between different inputs or tasks. The overall diagram of the FST module is shown in Figure 4. Here, feature maps extracted from RecBlock are treated as key ($K$) and value ($V$), and from SRBlock as query ($Q$).

First, feature maps extracted from RecBlock as key ($K$) and value ($V$) and from SRBlock as query ($Q$) are sent to the FST module, respectively. Then, for each patch $q_i$ and $k_j$ of $Q$ and $K$, the relevance $r_{i,j}$ between these two patches is calculated with the normalized inner product as shown in Equation (1):

$$r_{i,j} = \left\langle \frac{q_i}{||q_i||}, \frac{k_j}{||k_j||} \right\rangle.$$ (1)

Next, Feature Transfer Attention is calculated using this $r_{i,j}$ to transfer the structural features of the text image. In order to transfer the feature with the most relevant position in $V$ for each query $q_i$, we generate an index map $F$ that represents the most relevant position of $Q$ and $K$. The definition is shown in the following Equation (2):

$$f_i = \arg \max_j (r_{i,j}) \qquad i \in [1, \frac{h}{2} \times \frac{w}{2}],$$ (2)

where $h$ and $w$ are the height and width of the input, and $f_i$ is the index representing the $i$-th element of the feature map from the SRBlock and the most relevant position of the feature map from the RecBlock. Then, the feature $T$ transferred from the feature map of the RecBlock is obtained using $f_i$ as the index and applying the index selection operation to $V$. Here, the Feature Transfer Attention map $T$ contains the structural features of the text image and is denoised.

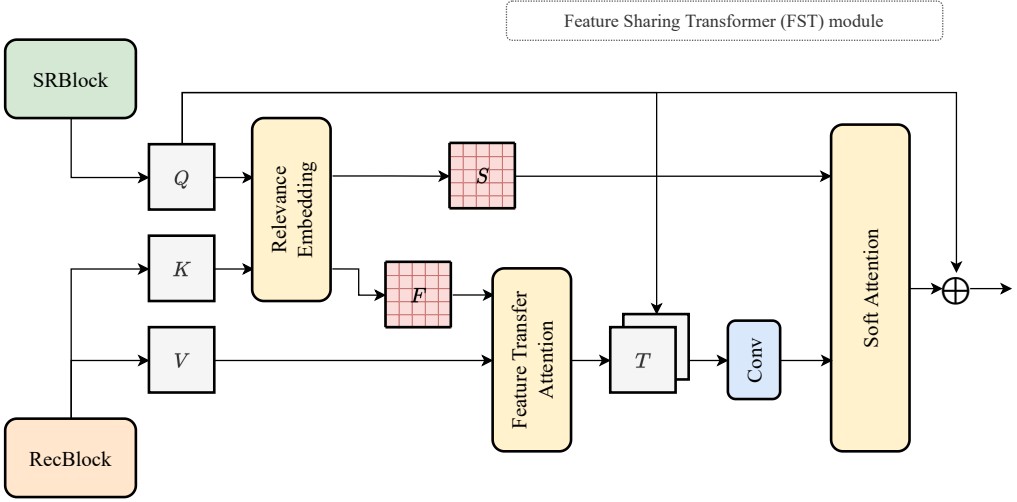

**Figure 4.** The architecture of the FST module. Feature maps extracted from the reconstruction branch are fused with the feature maps from the SR branch to utilize complementary characteristics, such as noise removal capability and text structure information.

Finally, Soft Attention is calculated to synthesize the features from $Q$ and Feature Transfer Attention map $T$. In order to transfer relevant features, a soft attention map $S$ is calculated from $r_{i,j}$ to estimate the confidence of the transferred texture features for each position of $T$ as shown in Equation (3):

$$s_i = \max_j(r_{i,j}), \tag{3}$$

where $s_i$ is the $i$-th position of the Soft Attention map $S$. Then, the feature map concatenating $T$ and $Q$ and the Soft Attention map $S$ are multiplied element by element and output as the final feature map $F_{out}$. The definition is shown in the following Equation (4):

$$F_{out} = Q \oplus Conv(Concat(Q, T)) \otimes S. \tag{4}$$

Here, $\otimes$ is the element-wise multiplication, and $\oplus$ is the element-wise summation.

### 3.3. Feature Enhancement Transformer (FET)

FET is a transformer designed to capture sequential information for arbitrary directions in text images tilted or curved using 2D positional encoding. BLSTM [16], which has been used to capture sequential information in our previous work [31] and TSRN [12], is effective for horizontal and vertical sequential information but is insufficient for text images of real scenes with various shapes and orientations. Therefore, inspired by the results of studies [15,27,28] on the text image using 2D attention and recent achievements of the transformer in computer vision [32–34], we attempt to use a transformer for a feature enhancement to capture sequential information in arbitrary orientations.

The attention mechanism in typical transformers used for natural language processing and time series data processing cannot recognize spatial location information or context because the input is processed in parallel in one dimension. Thus, positional encoding for 2D orientation considers spatial positional information to enable the transformer for text

images. Furthermore, we propose two transformers, one using 2D APE and the other using 2D RPE, and compare them in an ablation study.

### 3.3.1. 2D Absolute Positional Encoding (APE)

Unlike sequence data, text in images appears in various orders. In particular, text in real scenes often appears not only horizontally and vertically but also tilted or curved. Therefore, based on the feature maps, positional encoding is adaptively determined for the vertical and horizontal directions. For an index $p$ to the vertical and horizontal position of the input feature map, its positional encoding is defined as in Equations (5) and (6):

$$P_{p,2i} = sin(p/10000^{2i/D}),\tag{5}$$

$$P_{p,2i+1} = cos(p/10000^{2i/D}),\tag{6}$$

where $2i$ is the $2i$-th element in each axis $p$ of the positional encoding and $D$ is the number of dimensions in the depth direction. This is calculated for each of the vertical and horizontal directions. These positional encodings and feature maps are concatenated and flattened into a one-dimensional sequence, which is then input to the attention layer. Thus, the input $X$ in FET using APE is defined as in Equation (7):

$$X = Concat(X, P_h, P_w),\tag{7}$$

where $P_h$ is the vertical positional coding and $P_w$ is the horizontal positional coding. The architecture of the FET with 2D APE is shown in Figure 5.

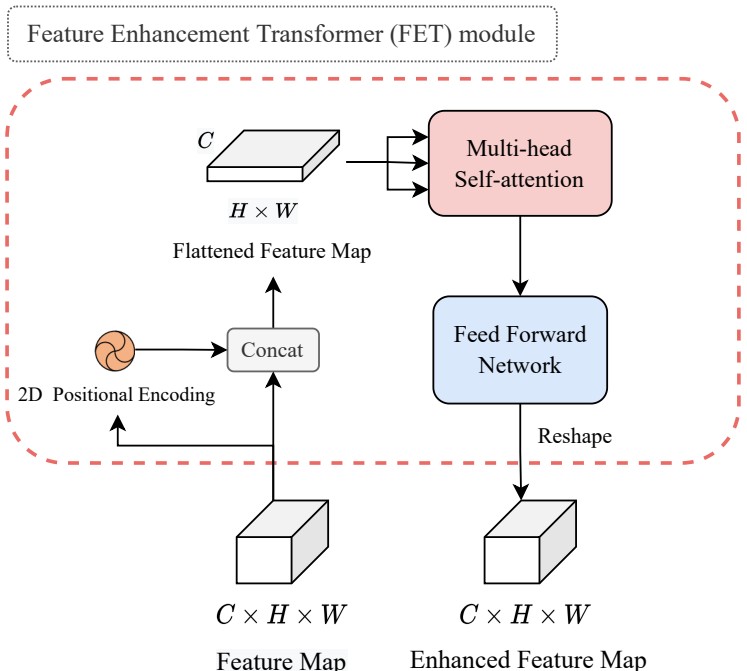

**Figure 5.** Illustration of FET module with self-attention modules using 2D absolute positional encoding.

In FET with 2D APE, for $n$ elements of input $x = (x_1, \ldots, x_n)$, the output $z = (z_1, \ldots, z_n)$ of self-attention is computed as a weighted sum of the input elements, as in the general transformer, as shown in Equation (8):

$$z_i = \sum_{j=1}^{n} \alpha_{ij}(x_j W^V),\tag{8}$$

where the projection $W^V$ is a parameter matrix and $\alpha_{ij}$ is each weight coefficient. $\alpha_{ij}$ is calculated using softmax and defined by Equation (9):

$$\alpha_{ij} = \frac{exp(e_{ij})}{\sum_{k=1}^{n} exp(e_{ik})}, \tag{9}$$

where $e_{ij}$ is the value calculated using scaled dot-product attention. It is defined in Equation (10):

$$e_{ij} = \frac{(x_i W^Q)(x_j W^K)^T}{\sqrt{d_z}}. \tag{10}$$

Here, the projections $W^Q$ and $W^K$ are parameter matrices and are unique for each layer. $d_z$ is the number of dimensions of $z$. In MHA, self-attention is calculated multiple times in parallel and then concatenated to produce the output. The output attention map is then sent to the Feed-Forward Network and finally reshaped to the same size as the input feature map.

3.3.2. 2D Relative Positional Encoding (RPE)

In recent studies [33,34], approaches that consider the relative position between each input element in the transformer have been proposed. Information such as the relative sequence and distance between elements is essential for tasks involving images. Following the RPE designed for images [33], we use a contextual RPE for text images in our FET to validate the effectiveness of the RPE. In RPE, encoding vectors are embedded in a self-attention module, and the relative positions between elements are trained during training for the transformer. The positional encoding vector $p_{ij}$ in 2D RPE is defined for Equation (11):

$$p_{ij} = (x_i W^K) rw_{ij}^T, \tag{11}$$

where $rw_{ij}$ is a trainable vector that denotes the relative positional weights between each position $i$ and $j$ of the input and interacts with the query embedding. In order to incorporate the encoding vector into the self-attention module, Equations (8) and (10) are reformulated as in Equations (12) and (13).

$$z_i = \sum_{j=1}^{n} \alpha_{ij}(x_j W^V + rw_{ij}^V). \tag{12}$$

$$e_{ij} = \frac{(x_i W^Q)(x_j W^K)^T + p_{ij}}{\sqrt{d_z}}, \tag{13}$$

Following [33], the $rw_{ij}$ is weighted by the Euclidean method based on the Euclidean distance between the elements and the Cross method, which considers the pixel position direction. These details are introduced in Appendix B. A self-attention module based on 2D RPE is shown in Figure 6. As in the case of using APE, the self-attention module is processed multiple times in parallel as the MHA, and then the output attention map is sent to the Feed-Forward Network.

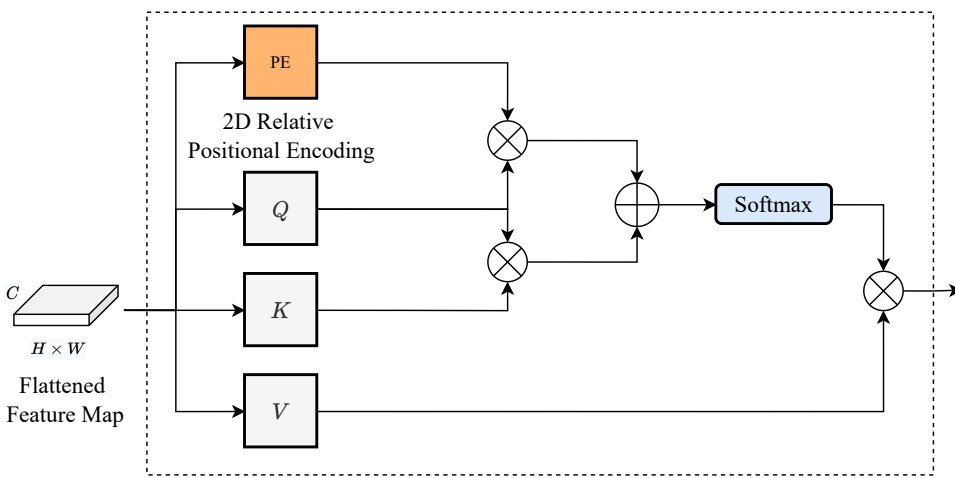

Modified self-attention for 2D relative position encoding

**Figure 6.** Illustration of self-attention modules with 2D relative position encoding.

*3.4. Loss Function*

In our MTSR, we attempt to optimize the model by minimizing the loss function for each SR and reconstruction branch, respectively. Following TSRN [12], we employed a combination of Gradient Profile Prior (GPP) [35] loss and L2 loss as the base loss function for each branch. The GPP loss $L_{GP}$ is defined as in Equation (14):

$$L_{GP}(x) = \mathbb{E}_x ||\nabla I_{gt}(x) - \nabla I(x)||_1 \quad (x \in [x_0, x_1]), \tag{14}$$

where $I_{gt}$ is the ground truth image, $I$ is the generated image, and $\nabla$ is the gradient fields. Then, $L_{sr}$ and $L_{rec}$, which are the loss functions for each branch, are defined as the sum of GPP loss and L2 loss, respectively. The final overall loss function $L$ is defined as in Equation (15):

$$L = \lambda_{sr} L_{sr} + \lambda_{rec} L_{rec}. \tag{15}$$

Here, $\lambda_{sr}$ and $\lambda_{rec}$ are arbitrary values and hyperparameters. The sum of $\lambda_{sr}$ and $\lambda_{rec}$ is set to a maximum value of 1.0.

**4. Experiment**

In this section, we first analyze the effectiveness of each module, such as FSTs and FETs, the effect of hyperparameters on the loss function, and the performance of two types of positional encoding, as ablation studies. Then, the performance of MTSR on the TextZoom dataset and its ability as a preprocessor in text recognition benchmarks are evaluated by comparing it with state-of-the-art methods. In all tables, the bolded value means the maximum number.

*4.1. Implementation Details*

All experiments are conducted on a PC with an Intel Core i7-9700k CPU and two NVIDIA GeForce RTX2070 super GPUs. Our MTSR consists of 8 blocks and uses a 4-heads MHA for FET. The model is implemented in Python and Pytorch and trained using Adam optimizer with a learning rate of $2 \times 10^{-4}$ on the TextZoom dataset. The batch size is 16, and the epoch is 300 in training. The images in the TextZoom dataset are text images culled from the SISR datasets RealSR [36] and SR-RAW [37]. These SISR datasets contain LR-HR pairs captured using digital cameras in real scenes. TextZoom consists of a training set of 17,367 images and a test set of 4373 images, with the test set divided into three subsets based on focal length: easy, medium, and hard. All input LR and target HR images are resized to $16 \times 64$ and $32 \times 128$, respectively.

The primary evaluation metric is the recognition accuracy for the generated images using the pre-trained text recognizers ASTER [21], MORAN [38], and CRNN [25]. In this process, the recognized text has all punctuation removed and uppercase letters converted to lowercase. As supplementary evaluation indices, we also use the Peak Signal-to-Noise Ratio (PSNR) and the Structural Similarity Index Measure (SSIM), which are general SR image quality evaluation indices.

### 4.2. Ablation Study

In this section, we will evaluate and analyze the effectiveness of each component for STISR, including each transformer module, the loss function, and the positional encoding. We conduct these experiments on TextZoom [12] and use the accuracy of pre-trained ASTER [21] for evaluation.

#### 4.2.1. Ablation Study on Each Module in the Backbone

We analyze the effect on accuracy with and without each module in the backbone to prove the effectiveness of BLSTM [16], FST, and FET for text images. Here, we use 2D APE for positional encoding in the FETs. The hyperparameters $\lambda_{sr}$ and $\lambda_{rec}$ of the loss function are set to 0.6 and 0.4, respectively. The comparison results are shown in Table 1. "Inference FPS" means the FPS of ASTER with the proposed model as preprocessing, and "Params" means the total number of network parameters. FPS is a metric for processing speed, the number of frames per second a model processes. Our previous study [31,39] corresponds to the configurations in row 5 of the table. The results in Table 1 show that the configurations using FET and FST, respectively, are more accurate than those using only BLSTM. It can further be seen that the configuration using all three modules achieves the highest accuracy. These results indicate that FET can capture sequential information in arbitrary directions that BLSTM cannot capture. We observe that FST can also transfer the noise-reduced structural features of the reconstruction branch. On the other hand, simultaneous use of these three modules inevitably increases the number of parameters and decreases the FPS. However, the FPS of 21.4 generally maintains real-time performance and balances accuracy and processing speed.

**Table 1.** Ablation study on each module. The comparison results on the accuracy of ASTER with and without each module in the backbone are shown. "Inference FPS" means the number of frames the ASTER can process per second with the proposed model as preprocessing, and "Params" means the total number of network parameters.

| BLSTM | FET | FST | Easy | Medium | Hard | Inference FPS | Params |
|:---:|:---:|:---:|:---:|:---:|:---:|:---:|:---:|
| ✓ | - | - | 74.3% | 56.6% | 38.8% | 25.3 | 3,072,972 |
| - | ✓ | - | 73.8% | 54.6% | 39.8% | 24.3 | 3,339,856 |
| - | - | ✓ | 72.1% | 55.6% | 39.5% | 23.2 | 3,568,116 |
| ✓ | ✓ | - | 74.2% | 57.6% | 40.9% | 22.7 | 3,935,732 |
| ✓ | - | ✓ | 74.1% | 57.9% | 40.3% | 22.1 | 4,075,088 |
| - | ✓ | ✓ | 73.5% | 55.9% | 40.8% | 21.7 | 4,202,616 |
| ✓ | ✓ | ✓ | **75.6%** | **59.8%** | **43.4%** | 21.4 | 4,937,848 |

#### 4.2.2. Ablation Study on Loss Function

To determine the best ratio of SR loss to reconstruction loss, we explore $\lambda_{sr}$ and $\lambda_{rec}$ from {0.0, 0.2, 0.4, 0.5, 0.6, 0.8, 1.0}. The comparison results are shown in Table 2. Here, we conduct experiments in a configuration with only the FST module to analyze which branch should be focused on in MTL. The results in the table show that settings of 0.4 to 0.6 for both $\lambda_{sr}$ and $\lambda_{rec}$ tend to have higher average accuracy. The average accuracy is lowest when either $\lambda_{sr}$ or $\lambda_{rec}$ is trained with a setting of 1.0 or 0.0, focusing on only one of the branches. These results indicate the effectiveness of balancing training in both SR and reconstruction branches.

**Table 2.** Ablation study on the loss function for MTL.

| $\lambda_{SR}$ | $\lambda_{Rec}$ | Easy | Medium | Hard |
|---|---|---|---|---|
| 0.0 | 1.0 | 70.78% | 55.07% | 38.05% |
| 0.2 | 0.8 | 71.02% | 55.12% | 38.13% |
| 0.4 | 0.6 | 71.15% | 55.75% | 39.14% |
| 0.5 | 0.5 | 71.07% | **56.06%** | 39.27% |
| 0.6 | 0.4 | 71.44% | 55.43% | **39.88%** |
| 0.8 | 0.2 | **72.02%** | 54.50% | 38.69% |
| 1.0 | 0.0 | 71.82% | 53.50% | 38.89% |

### 4.2.3. Ablation Study on Positional Encoding

In this section, we analyze the effects on the performance of two positional encoding approaches APE and RPE, for the FET module. Table 3 shows the comparison results. It can be seen that RPE can improve FPS with reduced parameters, while APE can improve accuracy. However, the effect is subtle. The results do not prove the significant effectiveness of RPE based on the Euclidean and Cross methods set up in this study.

Most previous studies [33,34] using RPE for computer vision tasks have indicated its effectiveness for tasks involving high-resolution images, such as object detection and image classification, and are not designed for low-resolution tasks, such as STISR. Therefore, the main reason for this experimental result may be that the model can not benefit from the relative positional information due to the tiny images handled in this task.

**Table 3.** Ablation study on the positional encoding approaches.

| PE Method | Easy | Medium | Hard | Inference FPS | Params |
|---|---|---|---|---|---|
| APE | 75.6% | **59.8%** | **43.4%** | 21.4 | 4,937,848 |
| RPE-Euclidean | 74.3% | 58.2% | 41.25% | 21.9 | 4,484,328 |
| RPE-Cross | **75.8%** | 58.6% | 42.30% | 22.2 | 4,370,253 |

### 4.3. Comparison with State-of-the-Art Methods

#### 4.3.1. Results on TextZoom

To prove the effectiveness of our MTSR, we compared it on TextZoom with seven other SR methods, including SRCNN [3], SRResNet [4], EDSR [29], LapSRN [40], TSRN [12], TSRGAN [14], and TBSRN [15]. Here, TSRN [12], TSRGAN [14], and TBSRN [15] are recent SR models specific to text images. As the configuration in this experiment, the positional encoding for FET, $\lambda_{sr}$, and $\lambda_{rec}$ are set to APE, 0.6 and 0.4, respectively. The results are shown in Table 4. It can be seen that our model achieves competitive performance with the current STISR methods, with some partially outperforming results. Especially at the medium and hard levels, the improvement in accuracy is comparatively significant, indicating that MTSR is effective for noisy and blurred images. TBSRN [15], a state-of-the-art method, uses a loss function with a pre-trained text recognition model for considering character position and recognition accuracy during learning. Benefiting from this learning method, TBSRN builds models specifically for text images and achieves high performance. On the other hand, although the loss function used to train our model is simple, it achieves comparable performance. Furthermore, our model significantly improves accuracy compared to EDSR [29], the baseline for the MTSR backbone. This result proves the effectiveness of MTL and the two transformers, and further performance improvement can be expected by specializing the learning method to text image as future work.

Comparison results of visual quality using PSNR and SSIM as complementary assessments are shown in Table 5. As with recognition accuracy, our model improves visual quality more for more challenging images such as medium and hard. However, the improvement in visual quality is less pronounced than in recognition accuracy compared to general-purpose SR methods. The reason is that PSNR and SSIM consider all image pixels,

and the recognition accuracy in Table 4 and visual quality in Table 5 are not necessarily proportional. Therefore, this is not an appropriate evaluation metric for STISR, which aims to reconstruct text regions.

**Table 4.** Comparison of recognition accuracy on the TextZoom dataset [12]. "Avg" indicates average accuracy in all subsets.

| Method | ASTER [21] | | | | MORAN [38] | | | | CRNN [25] | | | |
|---|---|---|---|---|---|---|---|---|---|---|---|---|
| | Easy | Medium | Hard | Avg | Easy | Medium | Hard | Avg | Easy | Medium | Hard | Avg |
| BICUBIC | 64.7% | 42.4% | 31.2% | 47.2% | 60.6% | 37.9% | 30.8% | 44.1% | 36.4% | 21.1% | 21.1% | 26.8% |
| SRCNN [3] | 69.4% | 43.4% | 32.2% | 49.5% | 63.2% | 39.0% | 30.2% | 45.3% | 38.7% | 21.6% | 20.9% | 27.7% |
| SRResNet [4] | 69.4% | 47.7% | 34.3% | 51.3% | 60.9% | 42.9% | 32.6% | 46.3% | 39.7% | 27.6% | 22.7% | 30.3% |
| EDSR [29] | 72.3% | 48.6% | 34.3% | 53.0% | 63.6% | 45.4% | 32.2% | 48.0% | 42.7% | 29.3% | 24.1% | 32.2% |
| LapSRN [40] | 71.5% | 48.6% | 35.2% | 53.0% | 64.6% | 44.0% | 32.2% | 48.3% | 46.1% | 27.9% | 23.6% | 33.2% |
| TSRN [12] | 75.1% | 56.3% | 40.1% | 58.3% | 70.1% | 53.3% | 37.9% | 54.8% | 52.5% | 38.2% | 31.4% | 41.4% |
| TSRGAN [14] | **75.7%** | 57.3% | 40.9% | 59.1% | 72.0% | 54.6% | 39.3% | 56.3% | 56.2% | 42.5% | 32.8% | 44.6% |
| TBSRN [15] | 75.6% | 59.5% | 41.7% | **59.4%** | **74.1%** | 57.0% | 40.8% | **58.4%** | **59.6%** | **47.1%** | 35.2% | **47.7%** |
| **MTSR (Ours)** | 75.6% | **59.8%** | **43.4%** | 58.9% | 73.9% | **57.2%** | **41.8%** | 56.0% | 56.2% | 47.0% | **35.3%** | 45.4% |

**Table 5.** Comparison of visual image quality based on PSNR and SSIM between the proposed method and representative STISR and SISR methods on the TextZoom dataset.

| Method | PSNR | | | SSIM | | |
|---|---|---|---|---|---|---|
| | Easy | Medium | Hard | Easy | Medium | Hard |
| BICUBIC | 22.35 | 18.98 | 19.39 | 0.7884 | 0.6254 | 0.6592 |
| SRCNN [3] | 23.13 | 19.57 | 19.56 | 0.8152 | 0.6425 | 0.6833 |
| SRResNet [4] | 20.65 | 18.90 | 19.53 | 0.8176 | 0.6324 | 0.7060 |
| EDSR [29] | 24.26 | 18.63 | 19.14 | 0.8633 | 0.6440 | 0.7108 |
| LapSRN [40] | 24.26 | 18.63 | 19.14 | 0.8633 | 0.6440 | 0.7108 |
| TSRN [12] | **25.07** | 18.86 | 19.71 | **0.8897** | 0.6676 | 0.7302 |
| TSRGAN [14] | 24.22 | 19.17 | **19.99** | 0.8791 | 0.6770 | 0.7420 |
| TBSRN [15] | 23.82 | 19.17 | 19.68 | 0.8660 | 0.6533 | **0.7490** |
| **MTSR (Ours)** | 23.55 | **19.88** | 19.64 | 0.8734 | **0.6843** | 0.7476 |

To evaluate the processing speed of the proposed model, we compare the inference FPS with and without super-resolution processing as preprocessing with the three methods EDSR [29], TSRN [12], and TBSRN [15]. The results are shown in Table 6. We can see that the speed effect of adding the proposed model to ASTER [21] as preprocessing is tiny and comparable in speed to the recent STISR method. For MORAN [38] and CRNN [25], adding MTSR results in a relatively lower FPS. However, our model is appropriate as a preprocessing method considering the balance with the improvement in accuracy. On the other hand, the proposed model tends to decrease the FPS compared to other STISR and SISR methods, although it maintains practicality. This is due to the increased computational cost caused by the multi-task architecture with two network branches in parallel, which is an issue that needs to be improved for the proposed model.

**Table 6.** Comparison of processing speed with and without SR as preprocessing.

| Method | Inference FPS | | | | |
|---|---|---|---|---|---|
| | Without Preprocess | EDSR [29] | TSRN [12] | TBSRN [15] | MTSR (Ours) |
| ASTER [21] | 24.3 | 24.2 | 24.2 | 19.4 | 21.4 |
| MORAN [38] | 124.4 | 99.8 | 110.9 | 72.8 | 72.2 |
| CRNN [25] | 765.1 | 208.6 | 348.7 | 168.0 | 143.1 |

4.3.2. Results on Scene Text Recognition Benchmarks

To evaluate the effectiveness of MTSR as a generalized STR preprocessor, we validate it using four STR benchmarks, including CUTE-80 [17], IIIT5K [18], SVTP [19], and SVT [20]. These four datasets contain text images with spatial deformations such as tilting or curving in real scenes. All 4580 images in the test set (288 from CUTE-80 [17], 3000 from IIIT5K [18], 645 from SVTP [19], and 647 from SVT [20]) in the four STR benchmarks were resized to $16 \times 64$ LR images. We further degrade the LR images by applying noise and blur processing to validate the effectiveness of our method. Specifically, Gaussian noise of $\sigma = 150$ was added to the original images for noise processing, and the original images were convolved with a $3 \times 3$ Gaussian kernel for blurring processing. Sample images are shown in Figure 7.

|  | CUTE 80 | IIIT5K | SVTP | SVT |
|---|---|---|---|---|
| Original | | | | |
| Gaussian Noise | | | | |
| Gaussian Blur | | | | |

**Figure 7.** Sample images from the STR benchmarks processed by Gaussian blur and Gaussian noise. Gaussian noise of $\sigma = 150$ is added to the original image as noise processing, and a $3 \times 3$ Gaussian kernel is convolved with the original image as blurring processing.

In this experiment, we evaluate our model in recognition accuracy using ASTER [21] and compare our model with EDSR [29], TSRN [12], and TBSRN [15]. The results are shown in Table 7. It can be seen that both degradation processes achieve better accuracy than the compared methods on most of the datasets.

**Table 7.** Comparison results on scene text recognition benchmarks including CUTE-80 [17], IIIT5K [18], SVTP [19], and SVT [20]. "GN" and "GB" refer to Gaussian noise and Gaussian blurring.

| Degradation | Method | Accuracy of ASTER [21] | | | |
|---|---|---|---|---|---|
| | | **CUTE-80 [17]** | **IIIT5k [18]** | **SVTP [19]** | **SVT [20]** |
| GN | EDSR [29] | 58.2% | 76.2% | 32.2% | 60.1% |
| | TSRN [12] | 60.7% | 78.0% | 36.9% | 61.5% |
| | TBSRN [15] | 63.5% | 80.2% | **39.0%** | **64.0%** |
| | **MTSR (Ours)** | **63.9%** | **80.4%** | 37.2% | 62.7% |
| GB | EDSR [29] | 59.2% | 51.5% | 20.1% | 62.15% |
| | TSRN [12] | 61.4% | 53.8% | 22.33% | 63.8% |
| | TBSRN [15] | 62.3% | 54.2% | **23.1%** | 65.1% |
| | **MTSR (Ours)** | **62.5%** | **54.6%** | 22.2% | **65.7%** |

*4.4. Limitations and Failure Cases*

In this section, we describe the limitations of our model found in the experiments. Figure 8 shows visualizations of our MTSR image generation failure cases in TextZoom [12]. We can see that text restoration by our MTSR is problematic in cases where characters

are indistinguishable from each other due to their proximity to each other and in cases of lengthy text. It is also difficult to recover text using our model for cases where the images have complex backgrounds and text parts are cut off in the image and have occlusions. The cases where unique fonts are used also pose difficulties to the model, and this is because the font patterns are not in the dataset and can be improved by extending the dataset.

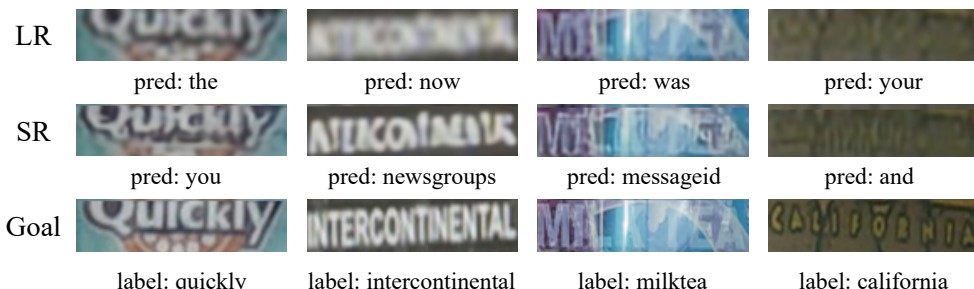

**Figure 8.** Examples of failures of images generated using MTSR. Several factors, such as complex backgrounds, unique fonts, and close spacing of characters, can limit text restoration using MTSR.

## 5. Conclusions and Discussions

In this paper, we proposed a Multi-task learning-based Text Super-resolution (MTSR) Network for scene text images. We adopt a multi-task architecture of SR and reconstruction, utilizing a transformer module called FST to recover text shape using complementary properties between the two tasks. Another transformer-based module, FET, was also used to capture sequential information for arbitrary directions by enhancing the feature map. Then, the feature maps output from each of these transformer modules was fused to improve the visual quality of text images of various shapes. Experimental results show that the proposed model achieves performance competitive with state-of-the-art methods on TextZoom. Furthermore, we proved the effectiveness of our model as a preprocessor not only for the STISR task but also for the STR benchmark.

However, our model does not achieve superior accuracy depending on the recognition method and test set, as shown in Table 4. One possible reason is that our learning methods, such as loss functions and label settings, are not sufficiently specialized for text images. It is also possible that our model does not sufficiently benefit from MTL due to the target image of the reconstruction branch being very close to the target image of the SR branch. Therefore, to build more text-specific models, our future work includes re-definition loss functions and the target image for the reconstruction branch.

**Author Contributions:** Methodology, K.H., M.K. and H.F.; Software, K.H.; Data curation, K.H.; Funding acquisition, H.F. and M.K.; Writing—original draft, K.H.; Writing—review and editing, M.K., H.F. and A.S. All authors have read and agreed to the published version of the manuscript.

**Funding:** This study is funded by JSPS/JAPAN KAKENHI (Grants-in-Aid for Scientific Research) #JP20K11955.

**Conflicts of Interest:** The authors declare no conflict of interest.

## Abbreviations

The following abbreviations are used in this manuscript:

| | |
|---|---|
| SR | Super-resolution |
| STR | Scene Text Recognition |
| LR | Low Resolution |
| SISR | Single Image Super-Resolution |
| HR | High Resolution |
| STISR | Scene Text Image Super-Resolution |

| BLSTM | Bidirectional LSTM |
| --- | --- |
| MTSR | Multi-task learning based Text SR |
| MTL | Multi Task Learning |
| FST | Feature Sharing Transformer |
| MHA | Multi-Head Attention |
| FET | Feature Enhancement Transformer |
| APE | Absolute Positional Encoding |
| RPE | Relative Positional Encoding |
| CNNs | Convolutional Neural Networks |
| RNN | Recurrent Neural Network |
| STN | Spatial Transformer Network |
| RecBlock | Reconstruction Block |

## Appendix A. Architectural Variations of the SRBlock

To investigate how to send the feature maps extracted from CNNs, considering the calculation of the two transformer modules FET and FST, we present three architectures of SR Block. Three architectural variations are shown in Figure A1: (a) CNNs and feature maps via BLSTM are input to FET and FST in parallel, (b) clean CNN feature maps without BLSTM are input to FET, and (c) FST and FET in series. Comparative results are shown in Table A1. The results show that architecture (b) improves accuracy by far. This result may be because the FET is designed to capture sequential information for arbitrary directions, and the sequential information for the horizontal and vertical directions of the BLSTM is noise to the FET.

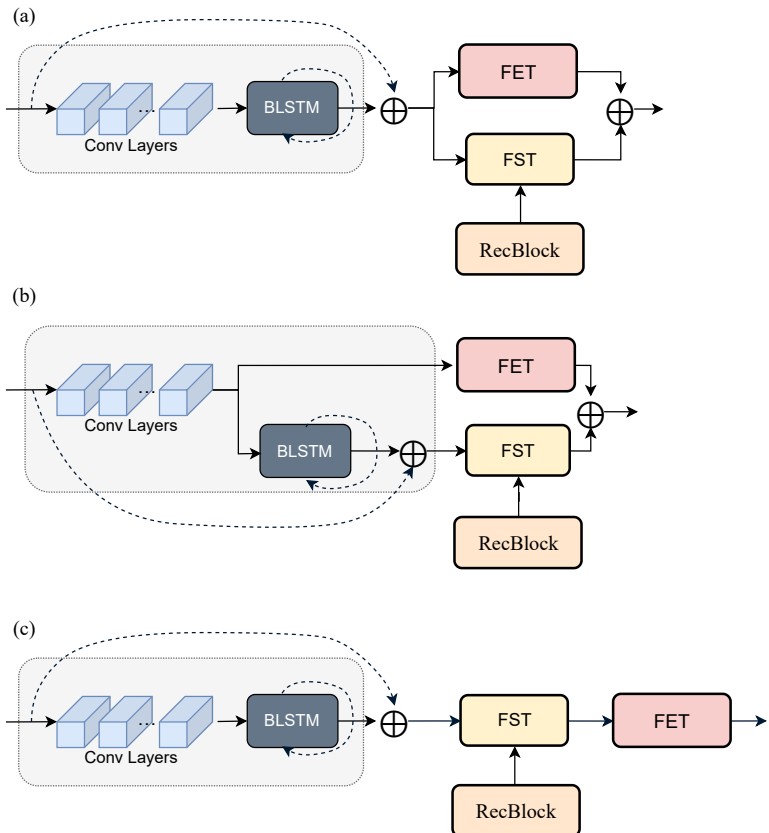

**Figure A1.** Three architectural of the SR Block: (**a**) CNNs and feature maps via BLSTM are input to FET and FST in parallel, (**b**) clean CNN feature maps without BLSTM are input to FET, and (**c**) FST and FET in series.

**Table A1.** Comparison results of the three SR Block architectures.

| Architecture | Accuracy of ASTER [21] | | |
|:---:|:---:|:---:|:---:|
| | Easy | Medium | Hard |
| (a) | 72.1% | 55.8% | 40.2% |
| (b) | **75.6%** | **59.6%** | **43.4%** |
| (c) | 73.9% | 54.5% | 40.9% |

**Appendix B. Descriptions on 2D Relative Position Calculation**

This section introduces how to calculate relative positions $rw_{ij}$ for 2D RPE in Section 3.3.2 of the main text. Both the Euclidean method and the Cross method are computed following [33].

**Euclidean method.** The relative positions $(\widetilde{x}_i - \widetilde{x}_j, \widetilde{y}_i - \widetilde{y}_j)$ on a 2D plane are defined as 2D coordinates, and the Euclidean distance between the two points is calculated. This distance is mapped to the corresponding encoding. This approach does not consider the orientation between elements. Equations (A1) and (A2) shows the definition:

$$rw_{ij} = P_{I(i,j)}, \tag{A1}$$

$$I(i,j) = g\left(\sqrt{(\widetilde{x}_i - \widetilde{x}_j)^2 + (\widetilde{y}_i - \widetilde{y}_j)^2}\right), \tag{A2}$$

where $P_{I(i,j)}$ is a learnable vector that stores the relative position weights. $I(i,j)$ is the 2D image of the target. The function $g(x)$ is a piecewise function based on [41] and indexed according to the relative position.

**Cross method.** This approach considers the position direction of the elements. Horizontal and vertical encodings are calculated separately, and they are fused by summarization. The method is given as Equations (A3)–(A5):

$$rw_{ij} = P^{\widetilde{x}}_{I^{\widetilde{x}}(i,j)} + P^{\widetilde{y}}_{I^{\widetilde{y}}(i,j)}, \tag{A3}$$

$$I^{\widetilde{x}}(i,j) = g(\widetilde{x}_i - \widetilde{x}_j), \tag{A4}$$

$$I^{\widetilde{y}}(i,j) = g(\widetilde{y}_i - \widetilde{y}_j), \tag{A5}$$

where $P^{\widetilde{x}}_{I^{\widetilde{x}}(i,j)}$ and $+P^{\widetilde{y}}_{I^{\widetilde{y}}(i,j)}$ are learnable vectors.

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
