# Peer review of "Multi-Task Learning for Scene Text Image Super-Resolution with Multiple Transformers"

_electronics, doi:10.3390/electronics11223813_

Round 1

Reviewer 1 Report

In this paper, the authors studied the Multi-task learning for scene text image super-resolution with multiple transformers. The results in this work are beneficial for various practical applications in the mid-IR region. The paper may be accepted for publication after getting the proper implementation of the following comments/suggestions.

1. In the abstract authors must define CNN.

2. In the introduction, the authors give an overview of studies addressing the issues of imaging and super resolution. From my perspective, the authors have recently missed developments in the field. To give the readers a comprehensive overview about this promising field I kindly suggest including the following references into the manuscript: Diouf, M., et al. Demonstration of speckle resistance using space–time light sheets. Sci Rep 12, 14064 (2022), and Mbaye Diouf et al "Multiphoton imaging using a quantitative CMOS camera", Proc. SPIE 11965, Multiphoton Microscopy in the Biomedical Sciences XXII, 119650D (2022).

3. What is the limitation of the MTSR network? I suggest that authors give more details of this method. 

 To my opinion, this manuscript can be recommended for publication in Electronics before the authors give the improvement.

Author Response

Reviewer 1: In this paper, the authors studied Multi-task learning for scene text image super-resolution with multiple transformers. The results of this work are beneficial for various practical applications in the mid-IR region. The paper may be accepted for publication after getting the proper implementation of the following comments/suggestions.
1. In the abstract authors must define CNN.
We corrected it on page 1, in blue color.
2. In the introduction, the authors give an overview of studies addressing the issues of imaging and super-resolution. From my perspective, the authors have recently missed developments in the field. To give the readers a comprehensive overview of this promising field I kindly suggest including the following references in the manuscript: Diouf, M., et al. Demonstration of speckle resistance using space-time light sheets. Sci Rep 12, 14064 (2022), and Mbaye Diouf et al "Multiphoton imaging using a quantitative CMOS camera", Proc. SPIE 11965, Multiphoton Microscopy in the Biomedical Sciences XXII, 119650D (2022).
We added the manuscripts of the imaging mentioned, as suggested in the introduction on page 1 and Reference [4] and [5], in blue color. Thank you for your suggestion of references.
3. What is the limitation of the MTSR network? I suggest that authors give more details about this method.
To my opinion, this manuscript can be recommended for publication in Electronics before the authors give the improvement.
We added the description of limitations on speed and performance of the proposed method on page 13 and page 14 in blue color, respectively.

Reviewer 2 Report

the authors proposed a multi-transformer based multi-task learning network for scene text image super-resolution. extensive ablation experiments confirm the effectiveness of each of the components of the proposed network. 

However, i have the following concerns for publication of this manuscript:

1.  the proposed network is not very competitive compared to SOTA methods, as evidenced by Table 4, Table 5, and Table 6.

2. How fast is the inference of the proposed network? i suggest to add corresponding runtime results compared to SOTA.

Author Response

Reviewer 2: The authors proposed a multi-transformer-based multi-task learning network for scene text image super-resolution. extensive ablation experiments confirm the effectiveness of each of the components of the proposed network.
However, I have the following concerns for the publication of this manuscript:
1. the proposed network is not very competitive compared to SOTA methods, as evidenced by Table 4, Table 5, and Table 6.
Yes, you are right, our model does not achieve significantly better performance than SOTA. But our model achieves performance comparable to or better than SOTA, and the use of multi-task learning and transformers significantly improves the base method (EDSR) performance, proving their effectiveness. We believe that these are novel and significant for future studies in this area. We also added an explanation about the differences between SOTA methods and our model, on pages 11 – 12, in blue color.
2. How fast is the inference of the proposed network? I suggest adding corresponding runtime results compared to SOTA.
We added the speed comparison results with SOTA in Table 6 on page 13. We also added an explanation of the speed of the proposed model on page 13, in blue color.

Round 2

Reviewer 1 Report

To my opinion, this manuscript can be recommended for publication in Electronics. The authors addressed all the comments and suggestions. 

Reviewer 2 Report

the authors addressed most of my concerns in my previoius review.

i have no more comments for publication of this manuscript.